# Bilateral Retinal Vein Occlusion-Simulated Coats’ Disease

**DOI:** 10.3390/diagnostics11050909

**Published:** 2021-05-19

**Authors:** Rui Hua, Meixia Zhang

**Affiliations:** 1Department of Ophthalmology, West China Hospital, Sichuan University, Chengdu 610041, China; woodshua@126.com; 2Department of Ophthalmology, First Hospital of China Medical University, Shenyang 110001, China; 3Research Laboratory of Macular Disease, West China Hospital, Sichuan University, Chengdu 610041, China

**Keywords:** retinal vein occlusion, Coats’ disease, optical coherence tomography, fluorescein angiography, anti-vascular endothelial growth factor therapy

## Abstract

Retinal vein occlusion (RVO) is a differential diagnosis for Coats’ disease due to retinal arterial Leber’s aneurysms. Occasionally, RVO shows a Coats-like appearance. The differential diagnosis between Coats’ disease and RVO is essential for clinical therapy, especially for those obsolete RVOs with collateral vessels and without retinal hemorrhage. In this case report, we describe and discuss the imaging characteristics of bilateral RVO-simulated Coats’ disease with tortuous retinal arterioles and its prognosis after anti-vascular endothelial growth factor therapy, which will be beneficial for its definite diagnosis and aid further investigation.

## 1. Introduction

Coats’ disease was first reported in 1908 as an ocular disorder with retinal telangiectasis and massive intraretinal and subretinal exudation [1]. Coats’ disease affects males more frequently than it does females and occurs unilaterally in more than 75% of cases [2]. Retinal vein occlusion (RVO) is a differential diagnosis of retinal arterial Leber’s aneurysms [3]. Moreover, capillary telangiectasia and retinal nonperfusion may develop in both RVO and Coats’ disease. RVOs occasionally exhibit a Coats-like appearance. In other words, a RVO can produce a capillary response simulating the characteristic features of Coats’ disease [4]. Distinguishing between Coats’ disease and RVO is essential for clinical therapy, especially in patients who have obsolete RVO with collateral vessels but no retinal hemorrhage. Previously, Luckie et al. reported the case of an adult patient with Coats’ disease secondary to a branch RVO in his left eye [5]. We present a case of bilateral branch RVO simulating Coats’ disease.

## 2. Case Presentation

A 52-year-old woman was presented to the ophthalmology department of First Hospital of China Medical University complaining of blurred vision in her left eye for two months. She had amedical history of hypertension and no diabetes. The best-corrected visual acuity was 60/60 in her right eye and 15/60 in her left, and the intraocular pressure was 17mmHg in both eyes. Slight cataracts were found in both eyes, and other anterior segment examinations were normal. Wide-field fundus imaging showed bilateral tortuous retinal arterioles with retinal hemorrhage, as well as tortuous and dilated retinal veins in the supratemporal quadrant. Heavy hard exudates were found in the macula of the left eye. The middle phase of fluorescein angiography revealed numerous miliary aneurysms in both eyes and patchy, blocked fluorescence in the left eye. Peripheral fluorescein angiography demonstrated the “fishing net” appearance of retinal capillaries and non-perfusion regions in both eyes (Figure 1). Multicolor imaging of the left eye highlighted macular stellar exudation and miliary aneurysms. Optical coherence tomography (OCT) showed a diffuse and cystoid macular oedema with hard exudation and subretinal fluid. OCT angiography of both eyes demonstrated miliary aneurysms and tortuous retinal arterioles (Figure 2). The diagnosis of bilateral RVO with tortuous retinal arterioles and secondary macular oedema in her left eye was made. She received one injection containing 0.05 mg ranibizumab for anti-vascular endothelial growth factor therapy, and1 month later the central foveal thickness on OCT decreased from 458 μm to 225 μm, with complete absorption of the subretinal fluid. The study adhered to the tenets of the Declaration of Helsinki and was approved by the Medical Research Ethics Committee of First Hospital of China Medical University. Informed consent was obtained from this patient.

## 3. Discussion

Herein, we described a bilateral branch RVO with a Coats-like appearance and tortuous retinal arterioles. Hypertension may be a risk factorfor RVO. In contrast to reports by Scimecaetal^4^ and Schatz et al [6], no secondary retinal detachment was found in our case. Unlike idiopathic perifoveal telangiectasia [7], which is an acquired bilateral neurodegenerative macular disease that usually manifests itself during the fourth to sixth decades of life, this patient had peripheral retinal lesions, without graying of the macula, right-angled retinal venules, refractile deposits in the superficial retina, hyperplasia of the retinal pigment epithelium, or foveal atrophy.

Familial retinal arteriolar tortuosity is characterized by tortuosity of the second- and third-order retinal arterioles [8]. Khan et al. reported a case involving tortuosity of the small and medium arterioles primarily in the macular region in both eyes, without vascular occlusion, or ischemia on fluorescein angiography [9]. By contrast, the co-existence of tortuous retinal arterioles and RVO exhibiting a Coats-like appearance has rarely been reported. In our case, Coats-like lesions occurred within the region of branch RVO, and retinal arteriolar tortuosity was present throughout the entire retina. Macular oedema in patients with RVO is treated with anti-vascular endothelial growth factor injections, which maintain retinal perfusion to achieve better visual outcomes [10].

In conclusion, this was the first report of bilateral RVO-simulated Coats’ disease with tortuous retinal arterioles. A definitive diagnosis is crucial for further investigation and treatment.

## Figures and Tables

**Figure 1 diagnostics-11-00909-f001:**
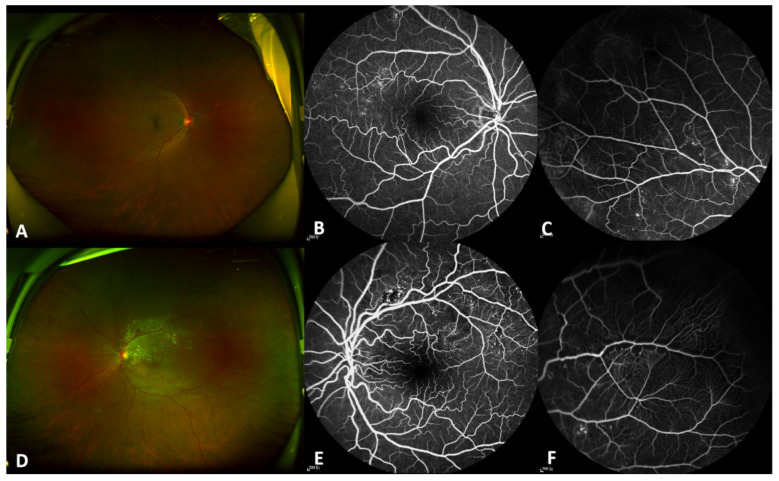
Widefield fundus images and fluorescein angiography of this patient at the initial visit. Widefield fundus images showing bilateral tortuous retinal arterioles (**A**, right eye; **D**, left eye) with retinal hemorrhage and tortuous and dilated retinal veins in the supratemporal quadrant. Heavy hard exudates were present in the macula of the left eye (**D**). The middle phase of fluorescein angiography revealed numerous miliary aneurysms in both eyes (**B**, right eye; **E**, left eye) and patchy blocked fluorescence in the left eye (**E**). Peripheral fluorescein angiography demonstratingthe “fishing net” appearance of retinal capillaries and nonperfusion region in the right (**C**) and left eyes (**F**).

**Figure 2 diagnostics-11-00909-f002:**
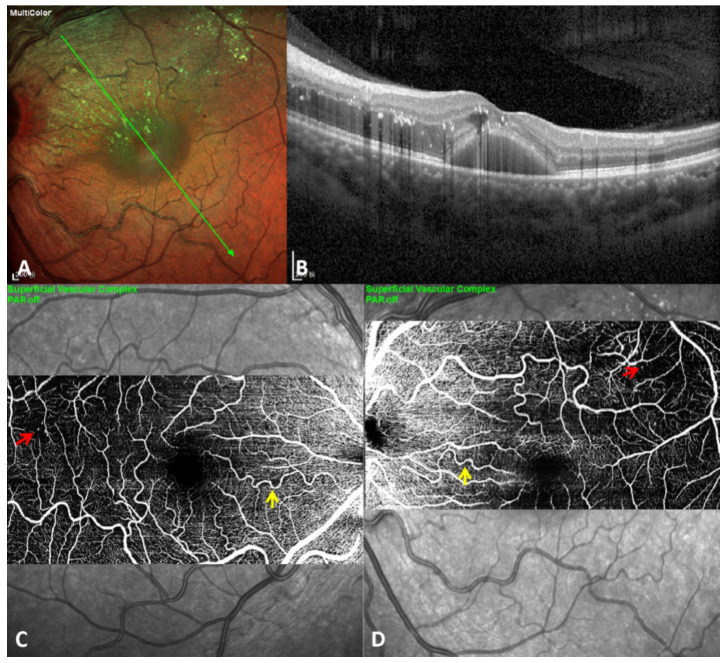
Multicolor imaging, optical coherence tomography, and optical coherence tomography angiography of the patient at the initial visit. Multicolor imaging of the left eye highlighted macular stellar exudation and miliary aneurysms (green arrow indicates the orientation of optical coherence tomographyin B) (**A**). Optical coherence tomography showed diffused and cystoids macular oedema with hard exudation and subretinal fluid (**B**). Optical coherence tomography angiography of the right eye revealed both miliary aneurysms (red arrow) and tortuous retinal arterioles (yellow arrow) (**C**). Optical coherence tomography angiography of the left eye revealed both miliary aneurysms (red arrow) and tortuous retinal arterioles (yellow arrow) (**D**).

## Data Availability

Authors declare that all related data are available in this manuscript.

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
