# Peer review of "Bilateral Retinal Vein Occlusion-Simulated Coats’ Disease"

_diagnostics, 2021, doi:10.3390/diagnostics11050909_

Round 1
Reviewer 1 Report
An interesting case study complements Luckie and hamilton's previous work (Adult Coats' disease in branch retinal vein occlusion) published in 1994. The plagiarism check did not reveal an overlap with previously published data. The reviewer suggests comparing the results herein with the ones of:
Scimeca G et al., Chronic exudtive ischemic superior temporal-branch retinal-vein obstruction simulating Coats's disease. Ann Ophthalmol 1986;18: 1 18-20.
Schatz H, Yannuzzi L, Stransky TJ. Retinal detachment secondary to branch vein occlusion: Part 1. Ann Ophthalmol 1976;8: 1437-52
This will give more insights into their work.
Author Response
- The reviewer suggests comparing the results herein with the ones of: Scimeca G et al., Chronic exudtive ischemic superior temporal-branch retinal-vein obstruction simulating Coats's disease. Ann Ophthalmol 1986;18: 1 18-20. Schatz H, Yannuzzi L, Stransky TJ. Retinal detachment secondary to branch vein occlusion: Part 1. Ann Ophthalmol 1976;8: 1437-1452.This will give more insights into their work.
Answer: We thankyou for this comment. We have compared our results to these references in the revised manuscript.
Reviewer 2 Report
In the manuscript authors describe RVO resembling Coats disease. The manuscript is poorly written, with numerous grammatical and punctuation errors. It looks like it has been written and never read after. In its merit, to my opinion it simply describes a case of potential subtle BRVO in a 52 year female. There is no information about crucial points like overall health, visual acuity, intraocular pressure, anterior segment of the eye. RVO is not an isolated ocular disorder, and therefore requires medical follow-up with RR measurement, cholesterol and clotting factors to the very least. Finally, the pictures provided does not resemble Coats disease, maybe its very early stages, and considering advanced age of the patient, it would be the last diagnosis to make.
Author Response
- In the manuscript authors describe RVO resembling Coats disease. The manuscript is poorly written, with numerous grammatical and punctuation errors. It looks like it has been written and never read after.
Answer: We thank you for this comment. This revised manuscript has been comprehensively edited by anative English speaker(Wolters Kluwer and Editage).
2.There is no information about crucial points like overall health, visual acuity, intraocular pressure, anterior segment of the eye.
Answer: We thankyou for this comment. We have added this information to the revised manuscript.
2.RVO is not an isolated ocular disorder, and therefore requires medical follow-up with RR measurement, cholesterol and clotting factors to the very least.
Answer: We thank you for this comment. We made a mistake and actually, this patient had a medical history of hypertension and no diabetes. Unfortunately, we did not carry on medical follow-up with RR measurement, cholesterol, and clotting factors. These examinations are not routine tests in our clinic for economic reasons. We apologise for this.
- The pictures provided does not resemble Coats disease, maybe its very early stages, and considering advanced age of the patient, it would be the last diagnosis to make.
Answer We thank you for this comment. This BRVO may resemble the early stage of Coats’ disease in the right eye and typical Coats’ disease in the left eye. All these lesions represent Coats’-like retinopathy. In addition, the age of this patient was within the age range for adult Coats’ disease. The hallmarks of Coats’ disease included unilateral peripheral retinal telangiectasis, aneurysm formation, and extensive subretinal exudation with serous retinal detachment, in the absence of other ocular or systemic vascular abnormalities (Luckie, A.P.; Hamilton, A.M. Adult Coats’ disease in branch retinal vein occlusion. Aust N Z J Ophthalmol 1994,22, 203-206.).Shields et al.also reported that the diagnosis of Coats’ disease relied on the presence of idiopathic retinal telangiectasia with intraretinal and/or subretinal exudation without retinal or vitreal traction (Shields, J.A.; Shields, C.L.; Honavar, S.G.; Demirci, H. Clinical variations and complications of coats disease in 150 cases: the 2000 Sanford Gifford memorial lecture. Am J Ophthalmol 2001, 131, 561–571). Some bilateral cases in older reports could represent secondary bilateral Coats-like retinopathy with systemic conditions (Ryan, S.; Schachat, A.; Sadda, S.; Wilkinson, C.; Hinton, D.; Wiedemann, P.Retina, 5th ed.; Saunders-Elsevier: London, England, 2013; pp.1059). Any patient with bilateral presumed Coats’ disease should be evaluated for conditions that cause Coats’-like exudative retinopathy (Vance, S.K.; Wald, K.J.; Sherman, J.; et al. Subclinical facioscapulohumeral muscular dystrophy masquerading as bilateral Coats disease in a woman. Arch Ophthalmol 2011, 129, 807-809). In our report, both hypertension and BRVO contributed to bilateral Coats’-like retinopathy.
Reviewer 3 Report
- Mention the underlying systemic diseases and work up for the etiology of RVO
- How you differentiated from Juxtafoveal retinal telangiectasia in right eye?
- Please mention if any other associated ocular findings.
- Where the case was presented? Name the Institute.
- Whether Institutional Review Committee approved the cases report?
- Mention about the informed consent of patient for case report.
Author Response
1.Mention the underlying systemic diseases and work up for the etiology of RVO.
Answer: We thank you for this comment. We apologise for a mistake and actually, this patient had a medical history of hypertension and no diabetes. Hypertension may be a risk factor for RVO.
2.How you differentiated from Juxtafoveal retinal telangiectasia in right eye?
Answer: We thank you for this comment. Unlike idiopathic perifoveal telangiectasia, which is an acquired bilateral neurodegenerative macular disease that usually manifests itself during the fourth to sixth decades of life, this patient had peripheral retinal lesions, without greying of the macula, right-angled retinal venules, refractile deposits in the superficial retina, hyperplasia of the retinal pigment epithelium, or foveal atrophy. We have added this information in the discussion section.
3.Please mention if any other associated ocular findings
Answer: We thank you for this comment. The best-corrected visual acuity was 60/60 in the patient’sright eye and 15/60 in her left. The intraocular pressure was 17 mmHg for her both eyes. Slight cataracts were found in both eyes, and other anterior segment examinations were normal. We have added this information in the case presentation section.
4.Where the case was presented? Name the Institute.
Answer: We thank you for this comment. This patient was admitted to the ophthalmology department of the First Hospital of China Medical University. We have added this information in the case presentation section.
5.Whether Institutional Review Committee approved the cases report?
Answer: We thank you for this comment. This case report was approved by the Medical Research Ethics Committee of the China Medical University. We have added this information in the case presentation section.
6.Mention about the informed consent of patient for case report.
Answer: We thank you for this comment. Informed consent was obtained from this patient. We have added this information in the case presentation section.
Round 2
Reviewer 2 Report
All my issues were adressed, although I still do not agree with the pattern in which this would be considered as a Coats disease. Looks like quite typical RVO.